# Relationship of Decrease in Frequency of Socialization to Daily Life, Social Life, and Physical Function in Community-Dwelling Adults Aged 60 and Over after the COVID-19 Pandemic

**DOI:** 10.3390/ijerph18052573

**Published:** 2021-03-04

**Authors:** Suguru Shimokihara, Michio Maruta, Yuma Hidaka, Yoshihiko Akasaki, Keiichiro Tokuda, Gwanghee Han, Yuriko Ikeda, Takayuki Tabira

**Affiliations:** 1Department of Rehabilitation, Medical Corporation, Nissyokai, Minamikagoshima Sakura Hospital, Kagoshima 890-0069, Japan; 2Doctoral Program of Clinical Neuropsychiatry, Graduate School of Health Science, Kagoshima University, Kagoshima 890-8544, Japan; m.maru0111@gmail.com; 3Department of Rehabilitation, Medical Corporation, Sanshukai, Okatsu Hospital, Kagoshima 890-0067, Japan; hidakayuma@icloud.com; 4Master’s Program of Health Sciences, Graduate School of Health Sciences, Kagoshima University, Kagoshima 890-8544, Japan; aka0805yfw1@yahoo.co.jp; 5Department of Rehabilitation, Tarumizu Chuo Hospital, Kagoshima 891-2124, Japan; 6Department of Rehabilitation, Medical Corporation, Gyokusyokai Kirameki Terrace Healthcare Hospital, Kagoshima 890-0051, Japan; gomyway.k.t@icloud.com; 7Department of Neuropsychiatry, Kumamoto University Hospital, Kumamoto 860-8556, Japan; hans11057@gmail.com; 8Department of Occupational Therapy, School of Health Sciences, Faculty of Medicine, Kagoshima University, Kagoshima 890-8544, Japan; yuriko@health.nop.kagoshima-u.ac.jp (Y.I.); tabitaka@health.nop.kagoshima-u.ac.jp (T.T.)

**Keywords:** COVID-19, community-dwelling adults 60 years or older, frequency of socialization, epidemiology

## Abstract

The study is cross-sectional in nature and aims to investigate the relationship of the frequency of socialization (FOS) to the daily life, social life, and physical function of community-dwelling adults aged 60 and over after the COVID-19 outbreak. A self-reported questionnaire survey was conducted on 3000 members of CO-OP Kagoshima, out of which 342 responses were received. Bivariate statistics was conducted followed by multiple logistic regression analysis. Questions with significant differences were set as independent variables, whereas the FOS was set as the dependent variable. Results indicate significant group differences between the decreased and increased/unchanged groups. After adjusting for potential covariates, multiple logistic regression analysis revealed decreases in the frequencies of cooking (OR: 0.07; 95% CI: 0.01–0.69; *p* = 0.02), shopping (OR: 18.76; 95% CI: 7.12–49.41; *p* < 0.01), and eating out (OR: 3.47; 95% CI: 1.21–9.97; *p* = 0.02), which were significantly associated with decreased FOS. The finding may inform policy making in identifying priorities for support in daily life for community-dwelling adults over the age of 60 undergoing social distancing.

## 1. Introduction

The number of COVID-19 cases in many countries around the world continues to increase since the first outbreak was reported in December 2019 [1]. The Japanese government declared a nationwide state of emergency on April 16, 2020, due to the rapid increase in the number of COVID-19 cases [2]. As such, public and recreational facilities were closed, and Japanese citizens were instructed to stay at home. Moreover, social distancing measures were implemented nationwide, thus requiring people to maintain a distance of approximately 2 m from one another and to avoid crowded places and unnecessary social gatherings. Despite the benefits, many studies reported that such a public health measure affected the physical and mental functions of older adults [3,4,5]. In Japan, the time allotted for physical activity among older adults decreased by approximately 30% during the state of emergency compared with before the spread of COVID-19, which prompted people to refrain from various activities [6]. Suzuki et al. [7] found that a decrease in physical activity before the COVID-19 pandemic was strongly associated with a decrease in subjective health among community-dwelling older adults in Japan. One of the reasons for this change may be the decreased frequency of socialization (FOS) among community-dwelling adults aged 60 and over due to social distancing measures. Based on the literature [8,9], the current study defined socialization as moving from one’s residence to another place or region. In this study, COVID-19-induced changes in the FOS were operationally defined as decreased or increased/unchanged FOS. Moreover, the FOS reflects the social activities of older adults, such as their roles inside and outside the home, leisure activities, and interactions with others [10]. Furthermore, Yasunaga et al. [11] used accelerometers to demonstrate that increased opportunities for socialization lead to more time for moderate physical activity among older adults, so an association with the FOS may be predicted.

The results suggest that increasing the FOS among community-dwelling older adults is an important factor in promoting physical and social activities and preventing confinement and frailty. However, studies that investigate the types of changes in daily life, social life, and physical factors that influence the FOS among community-dwelling adults over 60 years of age due to the COVID-19 pandemic are lacking.

To address this research gap, the present study aims to understand the changes in the FOS due to the COVID-19 pandemic among community-dwelling adults aged 60 and over in relation to changes in daily life, social life, and physical function. This form of intervention may contribute to the maintenance and improvement of the health of community-dwelling adults aged 60 and over whose socialization is restricted by COVID-19 measures.

## 2. Materials and Methods

### 2.1. Study Design

The study is cross-sectional in nature and employed a self-administered questionnaire.

### 2.2. Ethical Considerations

The participants were duly informed about the purpose of the study. Moreover, the researchers assured them that information collected will remain confidential and be used only for the study. By answering the questions in the questionnaire, the respondents indicated their agreement to participate. This study protocol was approved by the CO-OP Kagoshima Compliance Committee in September 2020 (Ref No. 7SR022-2009-00097).

### 2.3. Participants

A consumer’s co-operative (CO-OP) is a consumer organization whose common philosophy is to help people live prosperous lives beyond national borders [12]. The total number of union members across Japan is more than 25 million. At the local level, CO-OP Kagoshima is located in Kagoshima Prefecture, Japan. It is a private, community-based company that supports a range of activities, such as shop deliveries and home deliveries, to maintain its operation. As of 2018, it had approximately 310,000 members [13]. In September 2020, a self-administered questionnaire entitled Questionnaire on Changes in Lifestyle due to the COVID-19 outbreak was sent by post to 3000 randomly selected CO-OP Kagoshima members aged more than 20 years. In order to eliminate regional bias, we selected subjects evenly from municipalities in Kagoshima Prefecture. The questionnaire enclosed a reply envelope, such that responses were collected by post. The survey period lasted from September 2020 to November 2020, from which 1222 questionnaires were returned (response rate: 40.7%). In this study, one or more incomplete responses to questions about social demographics and baseline characteristics, health history, daily/social life, and physical health changes were excluded from the analysis. Out of the returned questionnaires, 759 respondents were aged 60 years and older. There were 417 questionnaires with missing data, which were excluded from analysis. Finally, 342 responses were analyzed in this study (Figure 1).

### 2.4. Questionnaire Structure

The questionnaire was developed by three skilled occupational therapists after content validation. A set of questions was developed based on the International Classification of Functioning, Disability, and Health (ICF) items with body structure, body function, activities, and participation as factors that influence changes in the FOS.

The questionnaire was presented with the following structure: (1) social demographics and baseline characteristics, (2) health history, (3) daily life changes, (4) social life changes, and (5) physical health changes. This questionnaire did not ask for identifiable personal information. Data were strictly controlled. By answering and returning the questionnaire, the respondents consented to participate in the study. The survey items in each section are outlined below.

#### 2.4.1. Social Demographics and Baseline Characteristics

This section included information on age, sex, height, family structure, and occupational status. The subjects were instructed to select the appropriate responses for the all questions.

#### 2.4.2. Health History

The subjects answered questions about underlying diseases (e.g., hypertension, diabetes, hyperlipidemia, hyperuricemia, osteoporosis, cancer, cardiovascular diseases, cerebrovascular diseases, Parkinson’s disease, depression, dementia, collagen diseases, spinal diseases, thyroid diseases, respiratory diseases, osteoarthritis, ophthalmic disorders, difficulty in hearing, fracture, and others). The subjects were asked to answer the items corresponding to their previous diseases in a multiple-response format. More than one answer could be provided. The number of responses was used as the number of underlying diseases.

#### 2.4.3. Daily Life Changes

This section inquired about changes in the FOS before (January 2020) and after the outbreak of the COVID-19 pandemic in Japan. We first asked, “How has your frequency of going out changed compared to before the COVID-19 pandemic?” Participants were asked to choose one of the following options: increased, decreased, and unchanged. Then, the respondents indicated whether a change was noted in the frequency or duration of activities related to daily living before and after the COVID-19 pandemic. The questionnaire consisted of a total of 15 items according to two categories.

The first included self-care and activities of daily living (ADL) with the following items: frequency of bathing, cooking, urination and defecation, hours of sleep, nap, bedtime, amount of food, and time of day to eat.

The second category focused on instrumental activities of daily living (IADL), which consisted of the following items: frequency of shopping, cleaning, laundry, number of phone calls, hours engaged in phone calls, amount of trash, and frequency of missing medicine.

Each question was answered using the following options: increased, decreased, or unchanged, for questions related to frequency; and longer, shorter, or unchanged, for questions related to duration.

#### 2.4.4. Social Life Changes

This section consisted of 10 questions according to the two following categories.

Questions under the first category were related to work and hobbies and consisted of the following items: time spent on hobbies and interests, roles and tasks at home (e.g., walking the dog, visiting a grave, and farming), commuting to work, and leisure (e.g., travel and spa).

The second category included interpersonal interaction and consisted of the following questions: opportunity to meet with friends and neighbors, time to talk to friends and neighbors, gatherings (e.g., neighborhood associations and senior citizen associations), family communication, eating out, and communication via the Internet (e.g., smart phones).

The questions were answered as follows: decreased, increased, and unchanged (or never done).

#### 2.4.5. Physical Health Changes

In this section, the following items were used to investigate physical health before and after the COVID-19 pandemic.

The first was related to weight changes. The participants had to choose one of three options, namely, gained, lost, or unchanged. The second pertained to changes in physical activity with three options, namely, increased, decreased, or unchanged. The third referred to one’s feeling of comfort with one’s body. The participants answered yes or no.

### 2.5. Statistical Analysis

The answers were tabulated by category and divided into two groups: decreased FOS and increased/unchanged FOS. A cross-tabulation table was created. The basic information and proportion of responses to each question were then compared. Student’s *t*-test was used for continuous variables, whereas the Mann–Whitney U test was used for ordinal scales. Moreover, Pearson’s chi-square test was used for categorical variables, whereas Fisher’s exact test was used to compare answers when the cross-tabulation table reached a point where more than 20% of the cells displayed the expected value of less than 5 [14,15]. Subsequently, residual analysis or multiple comparison was used as a post hoc test for items with significant differences across categories. In addition, non-linear logistic regression analyses were employed to examine the relationship of the FOS to daily life, social life, and physical factors that influence the FOS. Two regression models were used, namely, the crude and adjusted models. For each model, the FOS was set as the dependent variable. However, for the crude model, questions that exhibited significant differences across categories were individually set as independent variables. Meanwhile, the adjusted model adjusted for potential covariates, such as age, gender, family structure, occupation status, and underlying diseases. SPSS ver. 26.0 (IBM Corp., Armonk, NY, US) and R version 4.0.3 [16] were used for all analyses. *p* < 0.05 was considered statistically significant.

## 3. Results

### 3.1. Characteristics of the Participants

Table 1 presents the characteristics of the participants. Out of 342 adults aged 60 and over, 233 (68.1%) experienced decreased FOS. This group was younger than participants with increased/unchanged FOS (*p* = 0.016). No significant differences were observed in terms of other demographic factors between the two groups.

### 3.2. Comparison of Questionnaire Items by Frequency of Socialization

Table 2 displays the results of the cross-tabulation, bivariate comparison, and post hoc analysis according to the FOS. In terms of daily life changes, the increased/unchanged FOS group was more likely to report significant increases than the decreased FOS group in the following aspects: bathing, cooking, urination and defecation, frequency or duration of phone calls, and amount of trash. In addition, the decreased FOS group displayed a significant decrease in the frequency of shopping. In the increased/unchanged group, the following items were more frequently reported as unchanged before and after the COVID-19 pandemic: bathing, cooking, urination and defecation, shopping, frequency or duration of phone calls, and amount of trash.

Table 3 displays the results for social life changes. The following items were found to be more frequent in the decreased FOS group compared with the increased/unchanged FOS group: time spent on hobbies and interests, commuting to work, leisure, opportunity to meet with friends and neighbors, time for conversations with friends and neighbors, gatherings, family communication, eating out, and communication via the Internet. The decreased FOS group reported significant increases in the frequencies of time spent on hobbies and interests, roles and tasks at home, and communication via the Internet. Conversely, the increased/unchanged FOS group reported a significantly increased frequency of eating out. In addition, the increased/unchanged FOS group mentioned the same frequencies for time spent on hobbies and interests, roles and tasks at home, commuting to work, leisure, opportunity to meet with friends and neighbors, time for conversations with friends and neighbors, gatherings, family communication, eating out, and communication via the Internet before and after the COVID-19 pandemic.

Table 4 presents the results of physical health changes. In terms of weight, the decreased FOS group pointed to an increase compared with the increased/unchanged FOS group, and the decreased FOS group pointed to a significant decrease in physical activity. In addition, the increased/unchanged FOS group reported no significant changes in weight and physical activity.

### 3.3. Relationship of FOS to Daily Life, Social Life, and Physical Function

Table 5 provides the results of univariate and multivariate logistic regression analyses for both groups. Univariate logistic regression analyses indicated decreased frequencies for cooking meals (OR 0.09; 95% CI: 0.01–0.75; *p* = 0.03), shopping (OR: 17.48; 95% CI: 6.81–44.90; *p* < 0.01), family communication (OR: 2.27; 95% CI: 1.02–5.04; *p* = 0.04), and eating out (OR: 3.21; 95% CI: 1.11–9.26; *p* = 0.03), which were significantly related to the FOS according to the crude model. After adjusting for potential covariates, the decreased frequencies of cooking meals (OR: 0.07, 95% CI: 0.01–0.69; *p* = 0.02), shopping (OR: 18.76; 95% CI: 7.12–49.41; *p* < 0.01), and eating out (OR: 3.47; 95% CI: 1.21–9.97; *p* = 0.02) were significantly related to the FOS according to the adjusted model.

## 4. Discussion

The cross-sectional study conducted a questionnaire survey to understand changes in the FOS among community-dwelling adults aged 60 years and over following the declaration of the state of emergency in Japan as a response to the COVID-19 pandemic. The study investigated the relationship of the FOS to changes in daily life, social life, and physical function. The results reveal that decreased frequencies of cooking meals, shopping, and eating out were associated with the change in the FOS among community-dwelling adults aged 60 years or older even after adjusting for potential covariates. Thus, the results suggest that a set of daily living items should be prioritized to minimize the decline in the FOS among community-dwelling adults aged 60 and over.

The government prescribed self-isolation among older adults as a part of movement restraints given that older adults are more vulnerable with age. The willingness for pro-active self-isolation peaked among adults in the 70–75 year age group, which gradually increased among those aged 70 years [17]. Similarly, the results of the current study indicate that the decreased FOS group was significantly younger than the increased/unchanged FOS group. This finding suggests that positively refraining from socialization may have resulted in decreased FOS among the young older participants. Therefore, the study inferred that although adults aged 60 and over actively observed the measures for COVID-19 prevention, they continued to avoid densely populated areas, such as supermarkets.

The aspects of daily living include self-care, ADLs, and IADLs. The previous study [18] proposed that older adults maintained daily life at a minimal level due to COVID-19. In contrast, the present results reveal that the frequency for self-care remained nearly the same in the decreased and increased/unchanged FOS groups. Increases in the frequencies of bathing and elimination were observed only for the decreased FOS group. This finding is assumed related to the increased time spent at home [19]. The results reveal that FOS is significantly associated with frequencies of shopping and cooking. The reason for this finding may be that the majority of shopping activities involve socialization. Moreover, the study infers that less frequent cooking may indicate more time spent outside the home. Conversely, the restriction on eating out led to the assumption that they cook more frequently at home. In the decreased FOS group, longer times spent at home may result in more food to eat and rubbish to clean and more frequent and longer durations of telephone calls for social interaction. Furthermore, the high OR in the multiple linear logistic regression analysis of shopping frequency suggests that the majority of socialization for community-dwelling adults 60 years or older was allotted for shopping. Makino et al. [20] argued that limitations in outdoor IADLs were associated with the development of mild cognitive impairment (MCI), which points to the need for support of the maintenance of IADLs to prevent cognitive decline. In recent years, online services have enabled users to order daily necessities via the Internet, which can be delivered to their homes even before the decline in the frequency of shopping. Studies in Japan recently recognized that many older adults use the Internet for similar reasons, such as shopping and banking [21]. However, enhanced support for IADLs, including shopping, will be required in the future as people aged 60 years and above self-isolate to prevent COVID-19 infection.

Social factors included work, hobbies, and interpersonal interactions. The results demonstrate that the decreased FOS group reported significant declines in the majority of items related to social life. Interestingly, the proportion of people with increased time spent on hobbies and interests, roles and tasks at home, and communication via the Internet was significantly higher in the decreased FOS group. The reason underlying this result is that adults aged 60 and over in the decreased FOS group may spend more time at home due to the restrictions on socialization, thus lending more time for hobbies and tasks. In addition, they may use the Internet more frequently as an alternative means of interpersonal interaction. Alternatively, many adults aged 60 and over in the increased/unchanged group were originally engaged in hobbies or tasks at home. Moreover, the present study found that the frequency of eating out was significantly associated with decreased FOS. Thus, the study infers that family and friends typically accompany older adults when eating out, which is considered to provide opportunities for social interaction and support by promoting participation in shared social activities during meals [22]. In older adults, reduced social interaction due to the COVID-19 pandemic was associated with loneliness and depression [23]. In addition, community-dwelling older adults who were less satisfied with important activities were at high risk of exhibiting depressive symptoms [24], thus implying the importance of maintaining certain activities, such as hobbies and interpersonal interaction, despite the COVID-19 pandemic. Nevertheless, the need to take infection control measures, such as maintaining a distance of approximately 2 m [25] and wearing a mask [26], remains for activities that involve contact with people. Therefore, when providing social support for adults over the age of 60, implementing preventive measures is necessary in addition to hobbies and interpersonal interaction. The current situations require people to maintain physical distancing to prevent infection. However, Internet-based assistive technologies have been developed and reported to significantly reduce loneliness and increase social support and well-being in older adults [27]. Thus, their widespread use is expected.

The aspects of physical function pertain to weight, physical activity, and subjective health. The results indicate that the decreased FOS group reported significant weight gain and reduced physical activity compared with the increased/unchanged group. Di Santo et al. [28] found that 35% of community-dwelling older adults gained weight due to the COVID-19 lockdown, and such inactivity is a high-risk factor for impaired health and may lead to a state of sarcopenia, which is a combination of weight gain, loss of skeletal muscle mass, and muscle weakness [29]. Moreover, other studies demonstrated that psychological problems precede physical problems in older adults as a result of the COVID-19 pandemic [30,31]. In the present study, 21% of the participants reported weight gain, which is a slightly lower value than that in previous studies. Multiple logistic regression analyses suggested an association between an increase in physical activity and the FOS. However, no significant differences were identified (*p* = 0.06). Apart from demographic factors, such as being female and older age, the relationship of lifestyle factors such as increased consumption of sugary drinks and fried foods, increased frequencies of eating and snacking, decreased physical activity, and increased alcohol consumption to weight gain during the COVID-19 pandemic was identified [32]. Moreover, the results illustrate differences in the amount of activity dependent on the FOS. For example, 31% of the participants reported a decrease in physical activity, whereas only 5% increased their activity. Older adults who became more active during the COVID-19 pandemic reported an increase not only in certain activities, such as housework, but also in light exercises and sports, such as walking and flexibility exercises [7]. Thus, formulating a method for addressing the lack of physical activities among adults 60 years or older who refrain from socialization for COVID-19 prevention is necessary.

## 5. Limitations

The present study has its limitations. First, the subjects were CO-OP members aged 60 years or older and living in Kagoshima Prefecture. Moreover, the response rate of this questionnaire survey was somewhat low and subjects may be limited to adults’ age of over 60 years with relatively fair cognitive function and physical function that allows them to complete the questionnaire and post it to the mailbox. Thus, a possibility of selection bias exists, which limits the generalizability of the study. Second, changes in daily and social activities and physical function were self-reported and retrospective, which may led to recall bias. Third, confirming the causal relationship of the FOS to daily life, social activities, and physical functions is impossible due to the cross-sectional nature of the study. Thus, a longitudinal study should be conducted to confirm this relationship. Lastly, the need emerges to expand the scale of the study by recruiting participants from other districts.

## 6. Conclusions

The study found several differences between decreased or increased FOS and the daily and social activities and physical function of community-dwelling adults aged 60 and over. Among the participants, young older adults were more likely to report decreased FOS. This finding suggests the need to maintain certain levels of ADLs and IADLs even before the COVID-19 pandemic and the need to counteract weight gain and decreases in social or physical activities in adults 60 years or older in the decreased FOS group. Specifically, activities such as cooking, shopping, and eating out were significantly related to the FOS. The authors believe that the results can provide a reference and resources to aid in the daily lives of community-dwelling adults over the age of 60 during the COVID-19 pandemic.

## Figures and Tables

**Figure 1 ijerph-18-02573-f001:**
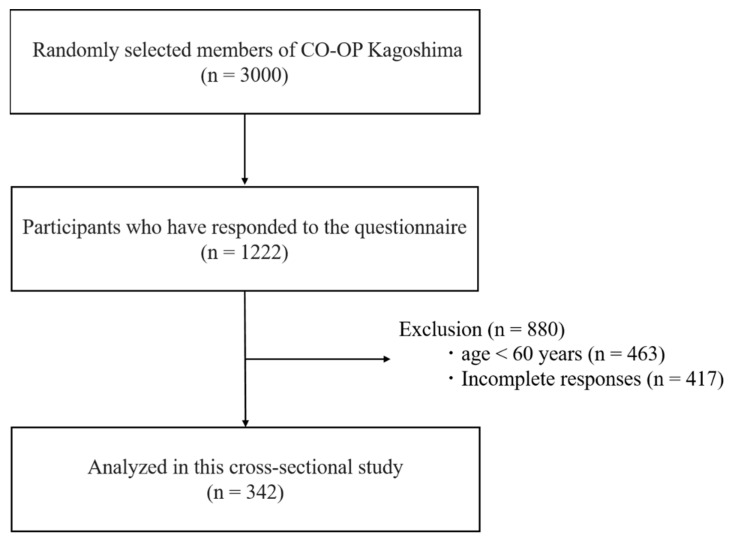
Flowchart for determining the participants.

**Table 1 ijerph-18-02573-t001:** Descriptive and bivariate statistics for participants.

	Overall, N = 342 ^1^	Increased/Unchanged FOS, *n* = 109 ^1^	Decreased FOS, *n* = 233 ^1^	*p*-Value ^2^
Age (years)	68.7 (7.2)	70.2 (7.8)	68.0 (6.8)	0.016 ^a^
Sex				0.57 ^b^
male	52 (15%)	18 (17%)	33 (14%)	
female	290 (85%)	91 (83%)	200 (86%)	
Family structure				0.37 ^b^
living alone	76 (22%)	21 (19%)	55 (22%)	
living with family	266 (78%)	88 (81%)	178 (78%)	
Height (cm)	154.7 (10.1)	154.5 (7.0)	154.8 (11.3)	0.8 ^a^
Occupation Status				0.7 ^c^
unemployed	7 (2.0%)	3 (2.8%)	4 (1.7%)	
remunerative employment	335 (98%)	106 (97%)	229 (98%)	
Number of underlying diseases	1.0 (0.0–5.0)	1.0 (0.0–4.0)	1.0 (0.0–5.0)	0.5 ^d^

^1^ Statistics presented: mean (SD); n/N (%); median (minimum–maximum); ^2^ statistical tests performed: (^a^) *t*-test; (^b^) chi-square test of independence; (^c^) Fisher’s exact test; (^d^) Mann–Whitney U test; FOS: frequency of socialization.

**Table 2 ijerph-18-02573-t002:** Bivariate statistics for daily life changes by frequency of socialization.

Daily Life Changes	Overall, N = 342 ^1^	Increased/Unchanged FOS, *n* = 109 ^1^	Decreased FOS, *n* = 233 ^1^	*p*-Value ^2^
Self-care and ADL				
Bathing				0.033 ^b^
increased	10 (2.9%)	0 (0%)	10 (4.3%) *	
decreased	3 (0.9%)	0 (0%)	3 (1.3%)	
unchanged	329 (96%)	109 (100%)	220 (94%) *	
Hours of sleep				0.084 ^a^
longer	18 (5.3%)	3 (2.8%)	15 (6.4%)	
shorter	35 (10%)	7 (6.4%)	28 (12%)	
unchanged	289 (85%)	99 (91%)	190 (82%)	
Bedtime				0.067 ^a^
longer	39 (11%)	7 (6.4%)	32 (14%)	
shorter	19 (5.6%)	4 (3.7%)	15 (6.4%)	
unchanged	284 (83%)	98 (90%)	186 (80%)	
Hours of nap				0.5 ^a^
longer	25 (7.3%)	6 (5.5%)	19 (8.2%)	
shorter	11 (3.2%)	2 (1.8%)	9 (3.9%)	
unchanged	306 (89%)	101 (93%)	205 (88%)	
Cooking				<0.001 ^b^
increased	49 (14%)	3 (2.8%)	46 (20%) *	
decreased	7 (2.0%)	3 (2.8%)	4 (1.7%) *	
unchanged	286 (84%)	103 (94%)	183 (79%) **	
The amount of food				0.4 ^a^
increased	29 (8.5%)	7 (6.4%)	22 (9.4%)	
decreased	22 (6.4%)	5 (4.6%)	17 (7.3%)	
unchanged	291 (85%)	97 (89%)	194 (83%)	
Time of day to eat				0.2 ^b^
longer	29 (8.5%)	6 (5.5%)	23 (9.9%)	
shorter	3 (0.9%)	0 (0%)	3 (1.3%)	
unchanged	310 (91%)	103 (94%)	207 (89%)	
Urination and defecation				0.012 ^b^
increased	21 (6.1%)	2 (1.8%)	19 (8.2%) *	
decreased	6 (1.8%)	0 (0%)	6 (2.6%)	
unchanged	315 (92%)	107 (98%)	208 (89%) *	
IADL				
Shopping				<0.001 ^b^
increased	2 (0.6%)	0 (0%)	2 (0.9%)	
decreased	168 (49%)	13 (12%)	155 (67%) **	
unchanged	172 (50%)	96 (88%)	76 (33%) **	
Cleaning				0.3 ^a^
increased	54 (16%)	12 (11%)	42 (18%)	
decreased	17 (5.0%)	6 (5.5%)	11 (4.7%)	
unchanged	271 (79%)	91 (83%)	180 (77%)	
Laundry				0.12 ^a^
increased	43 (13%)	9 (8.3%)	34 (15%)	
decreased	9 (2.6%)	1 (0.9%)	8 (3.4%)	
unchanged	290 (85%)	99 (91%)	191 (82%)	
Number of phone calls				0.001 ^a^
increased	48 (14%)	8 (7.3%)	40 (17%) *	
decreased	31 (9.1%)	4 (3.7%)	27 (12%) *	
unchanged	263 (77%)	97 (89%)	166 (71%) **	
Hours engaged in phone calls				0.044 ^a^
longer	43 (13%)	7 (6.4%)	36 (15%) *	
shorter	13 (3.8%)	3 (2.8%)	10 (4.3%)	
unchanged	286 (84%)	99 (91%)	187 (80%) *	
Amount of trash				0.023 ^a^
increased	54 (16%)	9 (8.3%)	45 (19%) *	
decreased	32 (9.4%)	9 (8.3%)	23 (9.9%)	
unchanged	256 (75%)	91 (83%)	165 (71%) **	
Missing medicine				0.8 ^b^
increased	14 (4.1%)	3 (2.8%)	11 (4.7%)	
decreased	9 (2.6%)	3 (2.8%)	6 (2.6%)	
unchanged	319 (93%)	103 (94%)	216 (93%)	

^1^ Statistics presented: mean (SD); n/N (%); ^2^ statistical tests performed: (^a^) chi-square test of independence; (^b^) Fisher’s exact test; * *p* < 0.05, ** *p* < 0.01 denote result of residual analysis or multiple comparisons; FOS: frequency of socialization; ADL: activities of daily living; IADL: instrumental activities of daily living.

**Table 3 ijerph-18-02573-t003:** Bivariate statistics for social life changes by frequency of socialization.

Social Life Changes	Overall, N = 342 ^1^	Increased/Unchanged FOS, *n* = 109 ^1^	Decreased FOS, *n* = 233 ^1^	*p*-Value ^2^
Work and Hobbies				
Time spent on hobbies and interests				<0.001 ^a^
increased	38 (11%)	4 (3.7%)	34 (15%) **	
decreased	161 (47%)	38 (35%)	123 (53%) **	
unchanged	143 (42%)	67 (61%)	76 (33%) **	
Roles and tasks at home				0.016 ^a^
increased	33 (9.6%)	5 (4.6%)	28 (12%) *	
decreased	46 (13%)	10 (9.2%)	36 (15%)	
unchanged	263 (77%)	94 (86%)	169 (73%) **	
Commuting to work				0.045 ^a^
increased	10 (2.9%)	3 (2.8%)	7 (3.0%)	
decreased	78 (23%)	16 (15%)	62 (27%) *	
unchanged	254 (74%)	90 (83%)	164 (70%) *	
Leisure				<0.001 ^b^
increased	5 (1.5%)	1 (0.9%)	4 (1.7%)	
decreased	281 (82%)	72 (66%)	209 (90%) **	
unchanged	56 (16%)	36 (33%)	20 (8.6%) **	
Interpersonal Interaction				
Opportunity to meet with friends and neighbors				<0.001 ^b^
increased	3 (0.9%)	2 (1.8%)	1 (0.4%)	
decreased	244 (71%)	52 (48%)	192 (82%) **	
unchanged	95 (28%)	55 (50%)	40 (17%) **	
Time to talk to friends and neighbors				<0.001 ^b^
increased	2 (0.6%)	1 (0.9%)	1 (0.4%)	
decreased	238 (70%)	53 (49%)	185 (79%) **	
unchanged	102 (30%)	55 (50%)	47 (20%) **	
Gatherings				<0.001 ^b^
increased	1 (0.3%)	1 (0.9%)	0 (0%)	
decreased	222 (65%)	55 (50%)	167 (72%) **	
unchanged	119 (35%)	53 (49%)	66 (28%) **	
Family communication				<0.001 ^a^
increased	21 (6.1%)	3 (2.8%)	18 (7.7%)	
decreased	128 (37%)	22 (20%)	106 (45%)	
unchanged	193 (56%)	84 (77%)	109 (47%)	
Eating out				<0.001 ^b^
increased	4 (1.2%)	3 (2.8%)	1 (0.4%) *	
decreased	279 (82%)	65 (60%)	214 (92%) **	
unchanged	59 (17%)	41 (38%)	18 (7.7%) **	
Communication via the Internet				<0.001 ^a^
increased	90 (26%)	21 (19%)	69 (30%) *	
decreased	80 (23%)	15 (14%)	65 (28%) **	
unchanged	172 (50%)	73 (67%)	99 (42%) **	

^1^ Statistics presented: mean (SD); *n*/N (%); ^2^ statistical tests performed: (^a^) chi-square test of independence; (^b^) Fisher’s exact test; * *p* < 0.05, ** *p* < 0.01 denote result of residual analysis or multiple comparisons; FOS: frequency of socialization; ADL: activities of daily living; IADL: instrumental activities of daily living.

**Table 4 ijerph-18-02573-t004:** Bivariate statistics for physical function changes by frequency of socialization.

Frequency of Socialization	Overall, N = 342 ^1^	Increased/Unchanged FOS, *n* = 109 ^1^	Decreased FOS, *n* = 233 ^1^	*p*-Value ^2^
Physical Health Changes				
Weight				0.010 ^a^
gained	74 (22%)	13 (12%)	61 (26%) **	
lost	29 (8.5%)	9 (8.3%)	20 (8.6%)	
unchanged	239 (70%)	87 (80%)	152 (65%) **	
Physical activity				0.002 ^a^
increased	17 (5.0%)	7 (6.4%)	10 (4.3%)	
decreased	106 (31%)	20 (18%)	86 (37%) **	
unchanged	219 (64%)	82 (75%)	137 (59%) **	
Uncomfortable with own body				0.4 ^a^
yes	87 (25%)	24 (22%)	63 (27%)	
no	255 (75%)	85 (78%)	170 (73%)	

^1^ Statistics presented: mean (SD); n/N (%); ^2^ statistical tests performed: (^a^) chi-square test of independence; * *p* < 0.05, ** *p* < 0.01 denote result of residual analysis; FOS: frequency of socialization; ADL: activities of daily living; IADL: instrumental activities of daily living.

**Table 5 ijerph-18-02573-t005:** Association between frequency of socialization and daily life, social life, and physical function.

	Crude Model				Adjusted Model			
	OR	95% CI		*p*-Value	OR	95% CI		*p*-Value
		Lower	Upper			Lower	Upper	
Daily Life Changes								
Bathing								
unchanged ^a^	-	-	-	-	-	-		
increased	11.6 × 10^7^	0.00		1.00	9.6 × 10^7^	0.00		1.00
decreased	2.9 × 10^7^	0.00		1.00	2.1 × 10^7^	0.00		1.00
Cooking								
unchanged ^a^	-	-	-	-	-	-		
increased	4.28	0.56	32.71	0.16	3.99	0.52	30.40	0.18
decreased	0.09	0.01	0.75	0.03	0.07	0.01	0.69	0.02
Urination and defecation								
unchanged ^a^	-	-	-	-	-	-		
increased	1.22	0.12	12.00	0.86	0.99	0.11	9.14	1.00
decreased	3.4 × 10^7^	0.00		1.00	2.1 ×10^7^	0.00		1.00
Shopping								
unchanged ^a^	-	-	-	-	-	-		
increased	12.8 × 10^8^	0.00		1.00	13.6 × 10^8^	0.00		1.00
decreased	17.48	6.81	44.90	<0.01	18.76	7.12	49.41	<0.01
Number of phone calls								
unchanged ^a^	-	-	-	-	-	-		
increased	0.69	0.15	3.12	0.63	0.72	0.16	3.27	0.67
decreased	1.53	0.32	7.32	0.60	1.61	0.32	8.21	0.57
Hours engaged in phone calls								
unchanged ^a^	-	-	-	-	-	-		
longer	1.33	0.23	7.71	0.75	1.13	0.19	6.86	0.89
shorter	0.56	0.09	3.63	0.54	0.49	0.06	3.74	0.49
Amount of trash								
unchanged ^a^	-	-	-	-	-	-		
increased	0.51	0.15	1.73	0.28	0.49	0.15	1.66	0.26
decreased	0.33	0.08	1.34	0.12	0.28	0.07	1.14	0.08
Social Life Changes								
Meet with friends and neighbors								
unchanged ^a^	-	-	-	-	-	-		
increased	0.00	0.00		1.00	0.00	0.00		1.00
decreased	1.92	0.62	5.96	0.26	1.83	0.58	5.80	0.30
Time to talk to friends and neighbors								
unchanged ^a^	-	-	-	-	-	-		
increased	12.8 × 10^7^	0.00		1.00	8.0 × 10^7^	0.00		1.00
decreased	0.90	0.30	2.73	0.85	1.05	0.33	3.30	0.93
Time spent on hobbies and interests								
unchanged ^a^	-	-	-	-	-	-		
increased	1.53	0.32	7.37	0.60	1.52	0.30	7.81	0.61
decreased	1.40	0.67	2.90	0.37	1.35	0.63	2.89	0.44
Roles and tasks at home								
unchanged ^a^	-	-	-	-	-	-		
increased	0.84	0.20	3.57	0.81	0.73	0.17	3.20	0.67
decreased	1.04	0.32	3.39	0.94	1.04	0.31	3.47	0.95
Gatherings								
unchanged ^a^	-	-	-	-	-	-		
increased	0.00	0.00	.	1.00	0.00	0.00		1.00
decreased	1.06	0.49	2.30	0.88	1.07	0.48	2.35	0.87
Family communication								
unchanged ^a^	-	-	-	-	-	-		
increased	5.24	0.55	50.08	0.15	5.52	0.52	58.19	0.16
decreased	2.27	1.02	5.04	0.04	2.18	0.97	4.93	0.06
Eating out								
unchanged ^a^	-	-	-	-	-	-		
increased	0.00	0.00		1.00	0.00	0.00		1.00
decreased	3.21	1.11	9.26	0.03	3.47	1.21	9.97	0.02
Leisure								
unchanged ^a^	-	-	-	-	-	-		
increased	6.5 × 10^7^	0.00		1.00	4.6 × 10^7^	0.00		1.00
decreased	0.77	0.25	2.32	0.64	0.80	0.25	2.57	0.70
Communication via the Internet								
unchanged ^a^	-	-	-	-	-	-		
increased	1.99	0.84	4.69	0.12	2.11	0.86	5.16	0.10
decreased	1.60	0.62	4.11	0.33	1.57	0.60	4.14	0.36
Commuting to work								
unchanged ^a^	-	-	-	-	-	-		
increased	0.21	0.01	3.78	0.29	0.27	0.01	5.24	0.38
decreased	0.61	0.22	1.68	0.34	0.62	0.22	1.77	0.37
Physical Health Changes								
Weight								
unchanged ^a^	-	-	-	-	-	-		
gained	1.18	0.42	3.26	0.75	1.11	0.39	3.15	0.84
lost	0.73	0.22	2.41	0.60	0.72	0.21	2.46	0.61
Physical activity								
unchanged ^a^	-	-	-	-	-	-		
increased	0.19	0.03	1.09	0.06	0.17	0.03	1.06	0.06
decreased	1.10	0.43	2.78	0.84	1.10	0.42	2.88	0.84

OR: odds ratio; CI: confidence interval; in each model, frequency of socialization set as a dependent variable; adjusted model: adjusted for age, gender, family structure, occupation status, underlying diseases; ^a^ denotes reference groups.

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
