# Peer review of "Relationship of Decrease in Frequency of Socialization to Daily Life, Social Life, and Physical Function in Community-Dwelling Adults Aged 60 and Over after the COVID-19 Pandemic"

_ijerph, 2021, doi:10.3390/ijerph18052573_

Round 1

Reviewer 1 Report

This is about the effect of COVID-19 on social activities which are very important for elderly....some questions are

  1.  the title says this is about older adults. I note that older adults in Japan is 65 or over.
  2. what is the response rate of older adults? how many older adults were mailed?
  3. low response rate and selection bias could be misleading the conclusion.
  4. what is definition of decrease in FOS?
  5. The result that decreases in the FOC are associated with activities of cooking, shopping, eating out is too much predictable. What do authors think about this?  

Author Response

Reviewer 1: This is about the effect of COVID-19 on social activities which are very important for elderly....some questions are

Responses to Reviewer #1: We appreciate your careful reading of our manuscript and your thoughtful suggestions to improve it. We have made revisions based on your comments and suggestions, and our point-by-point responses are given below. Revisions to the manuscript are indicated in red.

  1. the title says this is about older adults. I note that older adults in Japan is 65 or over.

Response1: Thank you for pointing this important information out. We have made changes in accordance with your comment. We have replaced the term [older adults] with [middle and older adults] in the title and text. Please confirm the revised manuscript.

  1. what is the response rate of older adults? how many older adults were mailed?

Response2: We thank the reviewer for this comment. As for the response rate of the questionnaire, the questionnaire used in this cross-sectional study was randomly selected by age and gender and mailed. Therefore, it is difficult to identify individual respondents. Thus, it was not possible to calculate the response rate limited to middle and older adults. In light of this, we have stated the response rate for the entire questionnaire (p.2, lines 93-94).

  1. low response rate and selection bias could be misleading the conclusion.

Response3: We thank you for pointing this important information out. As you pointed out, the subjects of this study included those who were able to complete and return the questionnaires, and those who were not able to complete and return the questionnaires may have been excluded. Therefore, the effect of selection bias has not been completely eliminated. Therefore, we add it as a limitation of our study (p.13, lines 329-331).

  1. what is definition of decrease in FOS?

Response4: We appreciate your comment. We defined decrease in FOS in this study as "a decrease in the frequency of socialization compared to before the COVID-19 pandemic." This was our operational definition based on previous studies by May, D et al. [1] and Baker, P.S et al. [2] Since the original manuscript did not fully describe the definition of decreased FOS, we have added an additional description in the revised manuscript (p.2, lines 57-59 and p.4, line 129-131).

References

  1. May, D. et al. The Life-Space Diary: A Measure of Mobility in Old People at Home. Int. Rehabil. Med. 1985, 7, 182–186, doi:10.3109/03790798509165993.
  2. Baker, P.S. et al. Measuring Life-Space Mobility in Community-Dwelling Older Adults. J. Am. Geriatr. Soc. 2003, 51, 1610–1614, doi:10.1046/j.1532-5415.2003.51512.x.

  1. The result that decreases in the FOC are associated with activities of cooking, shopping, eating out is too much predictable. What do authors think about this?

Response5: We thank the reviewer for this comment. Our study found several differences between decreased or increased/unchanged FOS and the daily and social activities and physical function of community-dwelling middle and older adults. Additionally, we found that specific activities such as cooking, shopping, and eating out were significantly associated with a decrease in FOS. This may suggest a risk factor for decreased FOS in middle and older adults, and may include a point of support for community-dwelling middle and older adults who are forced to live in the social distance by the Covid-19 pandemic. As you pointed out, this is a predictable result, but we think it has not been clarified that examines the association between FOS, daily/social activities, and physical function in middle and older adults living in the community.

Reviewer 2 Report

The article is interesting and pertinent. I recommend its publication with minor revisions. I suggest the following improvements:

- Explain any inclusion and exclusion criteria for selecting participants for the study;

- Clarify if procedures which ensure that the elderly have competence to answer the questionnaires were used;

- Clarify if any questionnaire validation procedure was carried out, to ensure the quality of information collected;

- Clarify the type of questions asked in the Health History domain;

- In the results description, I would suggest removing the % and p values from the text, in order to make it more comprehensible and easier to read.

Author Response

Reviewer 2: The article is interesting and pertinent. I recommend its publication with minor revisions. I suggest the following improvements:

Responses to Reviewer #2: We appreciate your careful reading of our manuscript and your thoughtful suggestions to improve it. We have made revisions based on your comments and suggestions, and our point-by-point responses are given below. Revisions to the manuscript are indicated in red.

- Explain any inclusion and exclusion criteria for selecting participants for the study;

Response: We appreciate your comment. The exclusion criteria for this study were those who had incomplete answers to questions about baseline characteristics, FOS, daily/social activities, and physical function. As you pointed out, the description of the exclusion criteria was inadequate, so we have added it to the revised manuscript (p.2-3, lines 95-98).

- Clarify if procedures which ensure that the elderly have competence to answer the questionnaires were used;

Response: We thank you for pointing this important information out. We did not use procedures to ensure that the subjects had the competence to answer the questionnaire in this regard. Therefore, there is a possibility of selection bias in this study. We have made the following statement about this point as a study limitation in the revised manuscript (p.13, lines 329-331).

- Clarify if any questionnaire validation procedure was carried out, to ensure the quality of information collected;

Response: Thank you for pointing this out. This questionnaire was developed by three skilled occupational therapists after content validation, and we believe that its validity is guaranteed. As you pointed out, we have clarified the validation procedure of the questionnaire in the revised manuscript (p.3, lines 104-105).

- Clarify the type of questions asked in the Health History domain;

Response: In the question in the Health History, the participants were asked to answer the items corresponding to their previous diseases in a multiple-response format. As you noted, we have added more detailed descriptions (p.3, lines 123-124).

- In the results description, I would suggest removing the % and p values from the text, in order to make it more comprehensible and easier to read.

Response: Thank you for this suggestion. We agree with you. We have revised this accordingly. Please confirm the revised manuscript.

Round 2

Reviewer 1 Report

I appreciate authors replied sincerely.

I would ask authors to add on;

  1. please describe the methods of mailing in detail. including how to select the mail list.  randomly selected by age and gender? e-mail or mail?
  2. please present the IRB-No.
  3. middle adult means usually 40+. So I would recommend "adults aged 60 and more"
  4. please describe more in discussion what kind of bias could be possibe due to selection bias(high non response). please describe it referring the general population(census) data of Kagoshima, for ex, the distribution of gender, family structure, employment..

Author Response

I appreciate authors replied sincerely.

I would ask authors to add on;

Responses to Reviewer #1: We appreciate your thoughtful suggestions to improve our manuscript again. We have made revisions based on your comments and suggestions, and our point-by-point responses are given below. Revisions to the manuscript are indicated in blue.

  1. please describe the methods of mailing in detail. including how to select the mail list. randomly selected by age and gender? e-mail or mail?

Response 1: We thank the reviewer for this comment. The questionnaire was sent by post to 3,000 randomly selected CO-OP Kagoshima members aged more than 20 years. However, in order to eliminate regional bias, we selected subjects evenly from municipalities in Kagoshima Prefecture. The questionnaire enclosed a reply envelope, such that responses were collected by post. As you pointed out, the manuscript needed a more detailed description of the procedure for conducting the survey. Please confirm the revised manuscript (p.2-3, lines 91-98).  

  1. please present the IRB-No.

Response 2: Thank you for pointing this out. The revised manuscript indicates the number for reference (p.2, line 84).

  1. middle adult means usually 40+. So I would recommend "adults aged 60 and more"

Response 3: We thank you for pointing this important information out. We have made changes in accordance with your comment. We have replaced the term [middle and older adults] with [adults aged 60 and more] or its equivalent in the title and text. Please confirm the revised manuscript.

  1. please describe more in discussion what kind of bias could be possible due to selection bias(high non response). please describe it referring the general population(census) data of Kagoshima, for ex, the distribution of gender, family structure, employment.

Response 4: We thank the reviewer for this comment. In light of your comments, we took another look at the response rate of the questionnaire. Referring to the data of Kagoshima prefecture, the percentage of the population over 60 years old in Kagoshima prefecture is 39.0% in 2019 [1]. On the other hand, 62.1% of the respondents in this study were aged 60 years or older (calculated from Figure 1). Therefore, we cannot eliminate the possibility that the respondents in this study were biased toward those aged 60 years or older although the respondents under 20 years old were not included in this study. Nevertheless, the percentage of members aged 60 years and above reported by the Japanese Consumers' Co-operative Union in 2018 was 48.7% in Japan [2], which we believe is a reasonable percentage considering that Kagoshima Prefecture has higher aging rates in Japan [3]. Also, when we compared the response rate of this survey (40.7%) with previous report that used a similar procedure to our study, the response rate was 31.0% [4], 48.3% [5], 42.5% [6], and 49.7% [7], which is undeniably somewhat low, but we think it is within a reasonable range. However, as you pointed out, the response rate is not high. Thus, as you pointed out, we could not eliminate the selection bias due to the low response rate, so we added it as a limitation of the study. We have added these points to the revised version of the manuscript because we needed to describe them in more detail in our manuscript (p.13, lines 331-334).

References

  1. Kagoshima Prefecture. Prefecture migration investigation http://www.pref.kagoshima.jp/ac09/tokei/bunya/jinko/jinkouidoutyousa/nennpou.html (ref. 2021/02/25)
  2. Japanese Consumers' Co-operative Union. news release https://jccu.coop/info/newsrelease/2018/20181116_01.html (ref. 2021/02/25)
  3. Kagoshima Prefecture. The population and household. http://www.pref.kagoshima.jp/ab13/kenko-fukushi/koreisya/koreika/kagoshimakennnokoureisyanogennzyounituite.html (ref. 2021/02/25)
  4. Ikeda Y, et al. Instrumental Activities of Daily Living: The Processes Involved in and Performance of These Activities by Japanese Community-Dwelling Older Adults with Subjective Memory Complaints. Int J Environ Res Public Health. 2019 Jul 23;16(14):2617. doi: 10.3390/ijerph16142617. PMID: 31340466; PMCID: PMC6678870.
  5. Takatori K, et al. The difference between self-perceived and chronological age in the elderly may correlate with general health, personality and the practice of good health behavior: A cross-sectional study. Arch Gerontol Geriatr. 2019 Jul-Aug;83:13-19. doi: 10.1016/j.archger.2019.03.009. Epub 2019 Mar 11. PMID: 30921602.
  6. Motohashi T, et al. [Examining attitudes toward mutual support in daily life and their associated factors within a community-based integrated care system: Findings of the "Survey to Enrich the Lives of Miyamae Ward Residents"]. Nihon Koshu Eisei Zasshi. 2020;67(3):191-210. Japanese. doi: 10.11236/jph.67.3_191. PMID: 32238755.
  7. Monma T, et al. Age and sex differences of risk factors of activity limitations in Japanese older adults. Geriatr Gerontol Int. 2016 Jun;16(6):670-8. doi: 10.1111/ggi.12533. Epub 2015 Jun 4. PMID: 26044713.